# Structured Framework and Genome Analysis of *Magnaporthe grisea* Inciting Pearl Millet Blast Disease Reveals Versatile Metabolic Pathways, Protein Families, and Virulence Factors

**DOI:** 10.3390/jof8060614

**Published:** 2022-06-09

**Authors:** Bhaskar Reddy, Sahil Mehta, Ganesan Prakash, Neelam Sheoran, Aundy Kumar

**Affiliations:** 1Division of Plant Pathology, Indian Council of Agricultural Research (ICAR)-Indian Agricultural Research Institute, New Delhi 110012, India; prakashg.ganesan@gmail.com (G.P.); haardikk@yahoo.co.in (N.S.); 2Crop Improvement Group, International Centre for Genetic Engineering and Biotechnology, New Delhi 110067, India; sahil.mehta@icgeb.res.in

**Keywords:** blast disease, *Magnaporthe*, sequencing, genome assembly, protein family, CAZymes, virulence, effectors

## Abstract

*Magnaporthe grisea* (T.T. Herbert) M.E. Barr is a major fungal phytopathogen that causes blast disease in cereals, resulting in economic losses worldwide. An in-depth understanding of the basis of virulence and ecological adaptation of *M. grisea* is vital for devising effective disease management strategies. Here, we aimed to determine the genomic basis of the pathogenicity and underlying biochemical pathways in *Magnaporthe* using the genome sequence of a pearl millet-infecting *M. grisea* PMg_Dl generated by dual NGS techniques, Illumina NextSeq 500 and PacBio RS II. The short and long nucleotide reads could be draft assembled in 341 contigs and showed a genome size of 47.89 Mb with the N50 value of 765.4 Kb. *Magnaporthe grisea* PMg_Dl showed an average nucleotide identity (ANI) of 86% and 98% with *M. oryzae* and *Pyricularia pennisetigena*, respectively. The gene-calling method revealed a total of 10,218 genes and 10,184 protein-coding sequences in the genome of PMg_Dl. InterProScan of predicted protein showed a distinct 3637 protein families and 695 superfamilies in the PMg_Dl genome. *In silico* virulence analysis revealed the presence of 51VFs and 539 CAZymes in the genome. The genomic regions for the biosynthesis of cellulolytic endo-glucanase and beta-glucosidase, as well as pectinolytic endo-polygalacturonase, pectin-esterase, and pectate-lyases (pectinolytic) were detected. Signaling pathways modulated by MAPK, PI3K-Akt, AMPK, and mTOR were also deciphered. Multicopy sequences suggestive of transposable elements such as Type LTR, LTR/Copia, LTR/Gypsy, DNA/TcMar-Fot1, and Type LINE were recorded. The genomic resource presented here will be of use in the development of molecular marker and diagnosis, population genetics, disease management, and molecular taxonomy, and also provide a genomic reference for ascomycetous genome investigations in the future.

## 1. Introduction

Due to the long history of cultivation, millets are considered “miracle grains” owing to their high nutritional quality, exceptional adaptability, and being a key source of food and nutritional security to developing nations [1]. Currently, millets are cultivated worldwide for food, feed, or forage resources, especially among low-income farmers [2,3]. Recently, millets have been recognized as “nutri-cereals” for production because they act as an excellent source of carbohydrates, proteins, dietary fiber, antioxidants, minerals, as well as vitamins [1,4]. Despite being climate-resilient crops in nature, epidemics of fungal diseases have been reported in millet crops, especially under intensive cultivation [5,6].

One such nutri-cereal that is affected by the blast disease is the pearl millet (*Pennisetum glaucum* (L.) R. Br.), a member of the family Poaceae [5,7,8]. The blast disease incited by filamentous ascomycete fungus, *Magnaporthe grisea* (T.T. Hebert) M.E. Barr (Anamorph: *Pyricularia grisea*), affects both forage and grain production [5,9,10]; this hemibiotrophic pathogen belongs to the *Magnaporthaceae* family and is of primary research interest owing to its agricultural adaptation, rapid dispersal, sustained dissemination, and its ability to cause crop loss in multiple monocots and consequent yield losses [11,12].

The blast disease was first reported in the year 1953 on a few pearl millet cultivars. Later, this disease disseminated sporadically to many varieties and hybrids during the 1980s [13]. From the year 2000 onwards, pearl millet blast disease was observed widespread across India [8,14]. Therefore, to prevent inadvertent dissemination and minimize the losses, the deployment of blast-resistant genes and the application of agrochemicals are widely practiced [8,14,15,16]. However, these strategies appear inadequate in blast-endemic areas because of variable host resistance, the emergence of new pathotypes, and the toxic nature of fungicides. Hence, the sustainability of pearl millet is becoming more challenging than ever before [2].

The pathogenic variability and population genetic analysis of *Magnaporthe grisea* infecting pearl millet have been attempted by many workers [17,18]. In the past decade (2010–2020), the rapid advancements in genomics and bioinformatics tools led to the reduction in sequencing cost and time [19]. Fungal genome sequencing has become affordable and enabled the determination of various genes and encoding proteins, associated pathways, virulence, and pathogenicity factors [20,21]. Recently, we reported detailed genomics contents of rice blast pathogen, *Magnaporthe oryzae* RMg_Dl strain [22].

Despite recent improvements in molecular techniques as well as sequencing, genomic loci associated with ecological adaptation, host invasion, tissue colonization, and blasting of the pearl millet host are not yet characterized in detail. Moreover, limited information describing the genome content for biological pathways and associated mechanisms for pearl millet blast fungal pathogen with spatial concern to cell wall invasion, signaling pathways, carbohydrate-active enzymes, and virulence factors are available. Therefore, to decipher the genomic potential and capability, we conducted whole-genome sequencing of pathogen *M. grisea* PMg_Dl-infected host *Pennisetum glaucum* (pearl millet) coupled with de novo genome assembly and functional annotation. The analysis reveals new insights on genomic contents, protein family, associated biochemical pathways, and their contribution to host–pathogen interaction. The present article describes the first report on the fundamental basis of genomic information and resource for future investigation on *M. grisea* infecting millets in general. Taken together, the genomic data will be beneficial to construct specific markers for pathogen surveillance, molecular classification, population genetics, and species delineation. It will ultimately be helpful to devise enhanced controlling strategies for pearl millet blast disease-causing pathogen.

## 2. Materials and Methods

### 2.1. Fungus Isolation and Host Range Susceptibility Study

In brief, the fungus *Magnaporthe grisea* PMg_Dl was isolated from peal millet cultivar ICMB95444 showing blast symptoms in the experimental farm located at the research farm of the Indian Agriculture Research Institute, New Delhi, India (Appendix A). Pathogenicity and host range assays of *Magnaporthe grisea* PMg_Dl were conducted on pearl millet (cv. ICMB 95444), finger millet (cv. Udar Malliga), wheat (cv. Agra Local), oats (cv. RD2035), and barley (cv. JHO 2000-4), as described previously [18]. The disease severity experiment on different host was conducted in a climate-controlled greenhouse (Rajdeep Agri. Products Pvt. Ltd., New Delhi, India); and the microscopic image of conidia were taken using ECLIPSE Ni-U upright fluorescence microscope with a NIS-Elements software, Japan (Nikon Imaging Japan, Inc., Tokyo, Japan).

### 2.2. Experimental Materials, Data Collection, Sequencing, and Assembly

In the current study, the pearl millet-infecting *Magnaporthe grisea* (Strain PMg_Dl) was genome-sequenced, curated, assembled, annotated, and published in GenBank with GenBank accession number RHLM00000000 (GenBank assembly accession GCA_003933175.1; with raw reads accession SRR8573217, SRR8573216, and SRR8776454). The sequences were subjected to the present analysis pipeline (Figure 1). The fungal genomic DNA was extracted and subjected to sequencing library preparation. The pair-end (PE) and mate-pair (MP) libraries were constructed using the Illumina TruSeq Nano DNA library prep kit and processed on an Illumina NextSeq 500 instrument with 2 × 150 bp chemistry. Additionally, a single-molecule real-time (SMRT) library was constructed and processed on the PacBio RS II platform with P6-C4 chemistry. The PE reads were quality-filtered using Trimmomatic v0.35 [23], and MP reads with NextClip [24]. The PE, MP, and long reads were assembled using SPAdes [25]; scaffolding was performed using SSAKE [26] and the final extension was performed using SSPACE version 3.0 [27].

### 2.3. Functional Genome Annotation of M. grisea PMg_Dl

Initially, the benchmark universal single-copy orthologs (BUSCO), v5.2.2, was utilized for genome assembly completeness evaluation against fungal ancestry [28]. The average nucleotide identity (ANI) among the genomes was obtained using OrthoANI [29]. The functional annotation of the genome was executed using the genome sequence annotation server (GenSAS)v6.0 [30] pipeline and the Galaxy platform [31]. RepeatMaskerv4.0.7 [32] was applied for repeat masking against fungi with GC content settings at 46–48%, and RepeatModelerv1.0.11 (http://www.repeatmasker.org/, accessed on 9 May 2022) was applied for repeat family determination. The genes and proteins were predicted using AUGUSTUSv3.3.1 [33]. The presence of tRNA and rRNA was determined using tRNAscan-SE v2.0 [34] and RNAmmerv1.2 [35], and simple sequence repeats (SSR) were determined with SSR Finderv1.0 [30]. The presence of common and unique orthologous proteins was obtained using OrthoVenn2 [36]. The presence of numerous protein families, superfamily, signal peptides, and gene ontology (GO) was determined using the InterProScan v5.48-83.0 [37]. Then, Bast2GO v5.2.5 [38] was utilized for hierarchical annotations and classification of the GO functional category. The dbCAN2 [39] database was utilized for the determination of carbohydrate-active enzymes (CAZymes) using the HMMER approach with an e-value of 0.00001. The presence of secretory proteins was analyzed against Fungal Secretome KnowledgeBase (FunSecKB) [40], and the peptidase proteins (proteolytic proteins) were determined against the MEROPS [41] database (release 12.4) with reference to *Magnaporthe grisea* sequences. The presence of effector proteins was determined through the initial identification of signal peptides using the Phobius tool among predicted total proteins [42]. Following this, the only signal peptide-featured sequences were subjected to EffectorP3.0 [43] for potential effector prediction, as described previously [22,44]. 

The presence of virulence factors (VFs) was determined using the database for virulence factors (DFVF) of sequences for rice blast [45] with the DIAMOND v0.9.26 [46] homology search tool with settings of ≥100 amino acid length, max-target sequence alignment 1, 60% subject coverage, 60% query coverage, 60% identity, and e-value of 0.00001. The presence of metabolic pathways was identified using the KEGG Automatic Annotation Server (KAAS) v2.1 web server [47] using the GHOSTX sequence homology search tool [48].

### 2.4. Phylogenetic Tree Analysis

The phylogenetic tree was generated using the alignment of 100 single-copy orthologous (SCO) proteins common among the 22 genomes (*M. grisea* PMg_Dl, *M. oryzae* 70-15, *M. oryzae* RMg-Dl, *Mag. poae* ATCC64411, *M. oryzae* WBKY1, *M. oryzae* B2, *M. oryzae* B71, *M. grisea* DS9461, *M. grisea* DS0505, *Magnaporthe* sp. MG03, *Magnaporthe* sp. MG05, *Magnaporthe* sp. MG07, *Magnaporthe* sp. MG08, *Magnaporthe* sp. MG12, *M. grisea* W98-20, *M. grisea* W97-11, *M. grisea* VO107, *M. grisea* DsLIZ, *M. grisea* NI907, *M. pennisetigena* Br36, *M. pennisetigena* PM1, and *Ophioceras dolichostomum* CBS). The 100 SCO proteins were aligned using a MAFFT v7.489 aligner [49]. The phylogenetic tree matrix was generated using the maximum parsimony approach with a setting of 1000 bootstrap and tree inference through subtree–prune–regraft (SPR), and then the phylogenetic tree was visualized in MEGA-X software [50].

## 3. Results

### 3.1. Genome Assembly, Assessment, and Genes Identification of M. grisea PMg_Dl

The hybrid de novo assembly of quality passed reads resulted in genome size of 47.89 Mb with a total of 341 scaffolds. The gene calling of the sequenced genome resulted in a total of 10,218 genes and 10,184 protein sequences (Table 1). The genome finish evaluation revealed that complete and single-copy orthologs were 744 (98.15%) out of a total of 758 orthologous BUSCO genes (Appendix A). The comparative genome average nucleotide identity (ANI) analysis revealed 86% similarity against *M. oryzae* and 98% against *Pyricularia pennisetigena* (Appendix A).

### 3.2. Phylogenetic Tree

For determination of PMg_Dl genome similarity, the phylogenetic tree was generated based on the concatenated 100 single-copy orthologous proteins among 20 *Magnaporthe* genera with *oryzae* and *grisea* species only, with the varying plant host (Figure 2, Appendix A). The constructed tree displayed that *M. grisea* PMg_Dl showed a close relationship with *M. grisea* and *M. pennisetigena* species infecting crabgrass and southern sandbur. Furthermore, we also observed that *M. grisea* PMg_Dl (host pearl millet) displayed a distant relationship with fungal species isolated from host finger millet, foxtail, and wheat plant. We also observed that fungal pathogens isolated from specific hosts were found clustered together, which indicates their close genetics (Figure 3).

### 3.3. Determination of Transposon and SSRs

A total of 18 unique transposon families were identified with a total of 18,020 copy numbers in this genome. Among the detected transposable family, the high copy numbers of Type LTR, LTR/Copia, LTR/Gypsy, DNA/TcMar-Fot1, and Type LINE were detected (Table 2). SSR analysis displayed 79,316 SSRs repeats with 243.24 Kb prevalence in the genome. Among them, di-nucleotide repeats were the most frequent, followed by trinucleotide repeats (Appendix A). In the case of dinucleotide repeats, the (TA)n was the most frequent, followed by (AT)n, (TC)n, and (AG)n. In the case of tri-nucleotide repeats, (CAG)n was the most frequent followed by (TGC)n and (GCC)n. Similarly, among tetra-nucleotides, the repeat (AATA)n was observed in high frequency, followed by (TTAT)n and (AAAT)n repeats.

### 3.4. Analysis of Orthologous Genes, Protein Family, and CAZymes in Assembled Genomes

The presence of orthologous genes determination in PMg_Dl and other genomes showed that 8631 (81.9%) genes were common among the studied genomes. The uniquely identified orthologous genes were 22, 1, 3, and 83 in genomes of PMg_Dl, DS9461, DS0505, and NI907, respectively (Figure 4A). Furthermore, the InterProScan was performed for the identification of protein family (PFAM) and superfamily, which displayed a total of different 3637 protein families, 695 superfamilies, and 446 PIRSF (Appendix A). The comparison of this genome with other genomes for the presence of common and unique PFAM showed that 3544 PFAM (94.3%) were shared among the studied genomes, whereas 29 PFAM (0.8%) were found uniquely in the PMg_Dl genome (Figure 4B). PFAM annotation displayed that family, namely the WD domain, was found to be highly expanded in the genome; the major facilitator superfamily, cytochrome P450, protein kinase domain, mitochondrial carrier protein, fungal-specific transcription factor domain, RNA recognition motif, and ABC transporter were also prevalent (Appendix A). Likewise, superfamily determination revealed that P-loop containing nucleoside triphosphate hydrolase, MFS transporter, alpha/beta hydrolase fold, NAD(P)-binding domain, protein kinase-like domain, and FAD/NAD(P)-binding domain were highly expanded in this genome (Appendix A). Furthermore, the majority of proteins were part of the transmembrane, followed by the non-cytoplasmic domain and cytoplasmic domain (Appendix A).

The CAZyme identification displayed a total of 539 CAZyme families, with high content of glycoside hydrolase (253 GH), auxiliary activities (110 AA), and glycosyltransferase (91 GT). The other detected families were carbohydrate esterase (50 CE) and carbohydrate-binding modules (11 CBM), and the least prevalent was pectin lyase (PL) (Table 3). Among the detected GH families, GH3, GH14, GH47, GH2, and GH76 were found relatively highly abundant. Among the AA family, AA7 (28) was the most abundant, followed by the AA9 (22) family. The families AA1, AA2, and AA3_2 were found equally prevalent with a total of nine families copied in this genome. In the CE family, CE5 was detected with the highest prevalence, followed by CE4 and CE1, and CE3 (Appendix A). Moreover, the comparative analyses showed that 168 (94.9%) CAZyme families were found shared among *M. grisea* PMg_Dl, DS9461, DS0505, and NI907 genomes. Additionally, six CAZymes, namely, AA1, GH109, GH35, GT109, GT61, and PL42, were found uniquely in the *M. grisea* PMg_Dl genome (Figure 4C).

### 3.5. Determination of Secretome and Peptidase Proteins

The predicted proteins’ secretome determination showed that 871 proteins were identified as secretory proteins. Furthermore, this protein classification at the protein family (PFAM) level revealed that these sequences contained 272 different PFAM, accounting for 777 total PFAM expanded in secretory proteins, in which families such as multicopper oxidase, subtilase family, glycosyl hydrolase family 61, GMC oxidoreductase, carboxylesterase family, and FAD-binding domain were highly expanded (Appendix A).

The peptidase featured proteins determination against the MEROPS database showed that a total of 163 proteolytic proteins were detected. These protein families’ content determination showed that different 120 families accounted for 270 PFAM. Among that family, the proteasome subunit, peptidase family M28, alpha/beta hydrolase fold, metallopeptidase family M24, peptidase M16 inactive domain, and insulinase (peptidase family M16) were highly prevalent (Appendix A).

To determine the detected peptidases and CAZymes for the secretory in nature, comparisons among secretome, MEROPS, and CAZyme mapped proteins were performed. The comparative Venn diagrams show that in the secretome, MEROPS, and CAZymes, the uniquely detected proteins were 586, 119, and 280, respectively (Figure 5A). Interestingly, there were 242 proteins (19%) shared among CAZymes and secretome, and 43 proteins (3.4%) among peptidase and secretome, which suggests that these proteins were secretory in nature and possibly destined to various organelles and extracellular environments and mediating diverse biological functions.

Moreover, the comparison of PFAM profiles between the secretome, peptidase, and CAZyme protein sequences showed that there were three shared PFAM, and 129, 80, and 111 PFAM were uniquely detected (Figure 5B). We also found various families shared among the secretome, peptidase, and CAZymes, suggesting the shared protein functionality and cooperation with each other for accomplishing diverse mechanisms. The unique families independently function and are possibly linked with their underlying mechanism.

### 3.6. Genome Functional Annotation Using Gene Ontology

The gene ontology (GO) annotation provides functional terminology and network interactions for gene products (proteins) in three classes: molecular function, biological process, and cellular components. GO-annotated sequences for biological processes ranged between 2709 and 7, molecular function ranged between 2786 and 5, and cellular components ranged between 1688 and 537 assigned sequences (Appendix A). The biological processes associated with most of the genes were metabolic processes, biological regulation, response to stimulus, signaling, and reproductive process (Appendix A). The molecular function associated with GO terms such as catalytic activity and binding were the two most highly prevalent categories (Appendix A). Similarly, the analysis revealed presence of high number of sequence counts in *M. grisea* PMg_Dl coding for cellular component associated with cellular anatomical entity (Appendix A).

### 3.7. Identification of Pathogenicity Genes, VFs, and Effectors

Blast analysis of genomic loci using the host–pathogen interaction database (PHI-base) revealed a total of 868 PHI-associated genes (Figure 6). We also found 51 virulence factors (VF) upon a homology search against DFVF that includes three VFs—GH17, GH37, and GH125 CAZymes—and two effectors each localized in plant cytoplasm and apoplast. There was only one VF classified to the peptidase subfamily M48A—a kind of metallo-endopeptidase. 

In this study, we deciphered 594 effectors among the signal peptides domain containing 1749 proteins. Among them, 338 (56.9%) were localized in the cytoplasm and 256 (43.1%) in the apoplast. Overall, the effector content in *the M. grisea* PMg_Dl proteome contributed 5.83% of total proteomes. Over 85.2% of the predicted putative effectors were small secreted proteins (<350 amino acids)—the feature of effector proteins.

Furthermore, the identified effectors were analyzed to determine the property of CAZymes; the results showed that 34 distinct families were mapped in a total of 70 CAZymes. AA (24 families) and GH (23 families) classes were equally present, with the highest occurrence of AA9 CAZyme (15 families), followed by CE5 (5 families), GH11 (4 families), and AA11, AA16, CE1, CE3, CE4, GH7 (3 families each) (Appendix A).

### 3.8. Identification of Metabolic Pathways in M. grisea PMg_Dl

KEGG pathway analysis revealed that a significant number of genes were found associated with metabolism (KO: 09100). In the carbohydrate metabolism (KO: 09101), fifteen different biochemical pathways were detected involving 277different enzymes (Table 4). Pathways such as glycolysis/gluconeogenesis, pentose and glucuronate interconversions, fructose and mannose metabolism, and starch and sucrose metabolism were also detected. 

We found cellulose degradation key enzyme EC 3.2.1.4 (endoglucanase), EC 3.2.1.91 (cellulose 1, 4-beta-cellobiosidase), and EC 3.2.2.21(beta-glucosidase) (Figure 7). Furthermore, pectin degradation enzymes, namely, EC 3.2.1.15 (endo-polygalacturonase), EC 3.1.1.11 (pectinesterase), and EC 4.2.2.2 (pectate lyases [polygalacturonate lyase]) of pentose and glucuronate interconversion pathways were also identified in this genome (Figure 8). Similarly, the secondary metabolite, terpenoid biosynthetic pathway mediated by the mevalonate pathway (acetyl-CoA to geranyl-PP and dimethylallyl-PP) and associated with 18different enzymes could be mapped (Figure 9). Additionally, environmental information processing (KO: 09130)-associated pathways such as signal transduction (KO: 09132) with 437enzymes were found involved in a total of 32 different signaling-associated biochemical pathways (Table 5). The signaling pathways such as Ras, MAPK, AMPK, PI3K-Akt, and mTOR with a significant number of enzymes for processing the information and underlying biological mechanisms were annotated.

## 4. Discussion

The advent of massively parallel sequencing technology with short-and long-read sequencing chemistries and the associated bioinformatics platform have altered the landscape of whole genome sequencing (WGS) worldwide. The recent reports on fungal genomes have enabled an in-depth understanding of metabolic pathways, pathogenicity/virulence factors, and protein families involved in host colonization and pathogenic adaptation. Here, we analyze and report the structured framework of the genome and metabolic pathways of pearl millet blast disease-inciting fungal pathogen *Magnaporthe grisea* PMg_Dl, which can be applied for another fungal genome study. In the sequenced genome, we identified 10,218 genes, with 10,184 of them being protein-coding sequences reported to be involved in the pathogenic life cycle, virulence, and host infection [51,52]. Additionally, the genome data offer an opportunity to decipher the evolution of gene(s) conferring fungicides resistance [53]. Mutation in these genes is presumed to have contributed to resistance against fungicides [53].

We observed the presence of diverse transposon families speculated to have links with pathogenicity and genome evolution [54,55]. Repeat-genomic regions (simple sequence repeats—SSR) are also a distinctive feature of fungal genomes; our data also indicated a high-frequency occurrence of dinucleotides followed by trinucleotides in the genome of *M. grisea* PMg_Dl. The detected SSRs could be exploited for fungal population assessment, pathogenic race monitoring, species/strain distinction, and strain level identification [56]. Such methods were successfully exploited for managing the leaf and fruit disease of citrus infected by the fungal pathogen [57].

In addition, the evolutionary relationship of pearl millet-infecting PMg_Dl with other *Magnaporthe* genomes was established using 100 single-copy orthologous proteins alignment-based phylogenetic tree. *Magnaporthe* strains infecting a particular host species clustered in the monophyletic clade is suggestive of a genetically homogeneous population; both *M. grisea* and *M. oryzae* are clustered separately, indicating the divergent evolution in monocot species, as earlier reported by Sheoran et al. [58]. Moreover, *M. grisea* PMg_Dl is genetically distinct from other *Magnaporthe* isolates infecting crabgrass, foxtail, and finger millets. The occurrence of diverse pathotypes of *Magnaporthe* is reflective of selection pressure from the host genotype [58]. The isolate *M. grisea* PMg_Dl showed varying degrees of blast severity on cereal hosts such as finger millet, wheat, barley, and oat in addition to the natural host pearl millet (Figure 3). Interestingly, rice is not infected by pearl millet-infecting *M. grisea,* indicating the infectivity preference of the isolate. A recent study on the 92 *Ascomycota* genome representing pathogenic and non-pathogenic fungi showed that ~80% of genes formed the core orthologous gene; the pathogenic group could be differentiated from the non-pathogenic group [59]. Additionally, the phylogenetic tree of orthologous genes depicted that pathogenic fungal isolates of the same lineage contain shared InterPro annotations (Figure 4).

The analysis of CAZyme families revealed a high representation of GH families, followed by AA and GT families in the genome. The CAZymes are reported to facilitate pathogen entry and colonization of the host by hydrolysis of host cell walls by degrading enzymes (CWDE) [60]. Various studies have shown that GH family proteins are involved in the degradation of complex wall-associated polysaccharides [21,22,60]. The catalytic activity of these protein families increases with the involvement of the CBM family and mediates the breakdown of the cell wall polysaccharides [61]. Furthermore, the genome analysis of phytopathogenic fungal species displayed heterogeneity for the CAZyme profile, suggesting diverse mechanisms of host invasion, infection process, and nutritional uptake [60,62]. Plant pathogenic fungi are reported to harbor diverse CAZymes compared to saprophytic fungi [60]. Among the pathogenic fungi, the hemibiotrophic pathogen *Magnaporthe* also showed a moderate quantum of secretome compared with biotrophic fungal pathogens [61,63]. Perhaps high content of complex polysaccharides in monocots appears to drive the evolution of diverse GH families in the pathogen infecting them [60,62].

Proteases play diverse roles in fungal physiological functions for nutrition, morphogenesis, and pathogenesis [64]. Peptidase family M28, zinc carboxypeptidase, subtilase family, and asparaginase are among the 120 peptidases protein families mapped in the *M. grisea* PMg_Dl genome. Carboxypeptidase is involved in the removal of the C-terminal residue of amino acids from dietary protein [65]. Subtilase has a wide range of peptidases, which are involved in acquiring nutrition through the breakdown of proteins into peptides and amino acids for fungal growth and development, and play an essential role in the host colonization [64,66]. Furthermore, the fungal peptidases have been implicated in altering or deactivating the host immune components, leading to immune suppression [67]. Hence, it is clear that peptidases are versatile for their diverse functions in a wide range of pH and temperature conditions, and are essential elements in adaptation to stress, fungal pathogenicity, and effector-mediated virulence and growth [64].

In the effector analysis, *M. grisea* PMg_Dl displayed a greater number of cytoplasmic effectors compared to apoplastic effectors. Additionally, the effector annotations revealed that the AA and GH families are highly represented, with the highest occurrence of the AA9 family. Similar results were also found in our previous report [22]. The cytoplasmic effectors migrate into the host cell via a biotrophic interfacial complex (BIC) that triggers pathogenesis events such as progressive secretion, hyphal elongation, and tissue colonization [68]. The apoplastic effectors act by spreading through the fungal cell wall and extra-invasive hyphal membrane (EIHM) [68]. In general, effectors are small secretory proteins that modify the host cell’s physiology, specific metabolite synthesis, and immunity function by suppressing PAMP-triggered immunity (PTI) [68,69]. Recent studies have confirmed that the host specificity [70] of pathogens is mediated by effector proteins, and is evolving through selection pressure from the host [71,72]. Furthermore, the polymorphism in effector proteins by mutations is reported for multi-host and expanded host–target binding [72,73]. The secreted proteins are recognized for limiting the defense signaling pathways to diminish the microbe-associated molecular pattern-triggered immunity. These secretomes also contain CAZymes, oxidoreductases, proteases, and lipases, which play a pivotal role in pathogen–host interaction [61]. Nonetheless, precise mechanisms of effector function, especially its host receptor evasion, and its role in disease syndrome mediated by effector-triggered susceptibility are poorly understood [71,72,73].

Additionally, we observed plant polysaccharide degradation pathways for starch, sucrose, pentose, and glucuronate. The coding sequences for starch- and sucrose-degrading hydrolytic endoglucanase (EC 3.2.1.4), beta-glucosidase (EC 3.2.2.21), and cellulose 1, 4-beta-cellobiosidase (EC 3.2.1.91) were found in the genome (Figure 7). The cellulolytic pathway mediates the conversion of cellulose into glucose, which serves as an essential substrate for energy production [74]. Another plant cell wall constituent is pectin. Genes for endo-polygalacturonase (EC 3.2.1.15), pectate lyases (EC 4.2.2.2), and endo-1, 4-beta-xylanase (EC 3.2.1.8) hydrolyzing pectin were detected in the genome (Figure 8) [75].

Moreover, comparative functional genome annotation among secretome–peptidase–CAZyme proteome revealed that 19% secretome–CAZyme and 3.4% secretome–peptidase proteome were shared. Similar results were also observed when comparatively annotated against PFAM, wherein 22.6% secretome–CAZyme and 17.2% secretome–peptidase PFAM were shared. This suggests that various CAZyme and peptidases are secretory in nature and associated with transboundary and diverse biological functions. Interestingly, none of the proteins was found to be shared among these three features, which indicates a substrate preference for their action (Figure 5A). We observed only three PFAM shared among secretome–peptidase–CAZyme proteome featured sequences, suggestive of the mechanism where two or more different protein families/domains joined together to perform the biological mechanism. Conversely, the detachment of a single domain altered the protein/enzyme functionality [76]. A similar observation was reported for *Colletotrichum truncatum*, the causal organism of chilli anthracnose disease. In one study, the authors affirmed the presence of various proteins shared among secretome–peptidase–CAZymes in the genome [77]. 

*M. grisea* PMg_Dl genome exhibited a diverse composition of protein families and superfamilies, which accounted for 3637 PFAM, 695 superfamilies, and 446 PIRSF in the genome. It is noted that the genome of *M. grisea* PMg_Dl consists of a high distribution of protein families, namely, WD domain, cytochrome P450, major facilitator superfamily, mitochondrial carrier protein, protein kinase domain, ABC transporter, and fungal-specific transcription factor domain. Similarly, the protein superfamily, P-loop-containing nucleoside triphosphate hydrolase, was the most widely distributed in the genome, followed by NAD(P)-binding domain superfamily, MFS transporter superfamily, protein kinase-like domain superfamily, and FAD/NAD(P)-binding domain superfamily. The WD domain is reported for involvement in various biochemical events such as nucleic acid replication, transcriptional control, impairment mechanisms, RNA maturation, signal transduction [78], growth, differentiation, and virulence [79]. The major facilitator superfamily of proteins acts for the transport of amino acids, simple sugars, oligosaccharides, antibiotics, and nucleotides through membranes in the bidirectional mode [80,81]. Such transporter proteins not only modulate the toxin-mediated pathogen virulence but are also believed to protect pathogens from plant-secreted toxins [80]. The identified protein families are reported to have functions such as MAPK for development [82] and pathogenicity/virulence [83,84] that generally facilitate pathogen–host interaction [85]. For example, we identified a secretome containing a conserved small secretory protein family, cerato-platanin, which is a kind of fungal pathogenicity mediator [86]. In *M. grisea* PMg_Dl, we found a single copy of a cerato-platanin domain that is conserved in 91 other fungal species under Pezizomycotina or Agricomycotina. Interestingly, loss of genes containing the cerato-platanin domain is also reported in Agricomycotina, wherein 23 gene duplications and 49 gene losses were observed in Agricomycotina. Moreover, the pathogenicity-related domains, namely the CFEM domain and ribonuclease/ribotoxin, were found in *M. grisea* PMg_Dl that play essential roles in the metabolic pathways [63].

Earlier studies confirmed that MAP kinase and cAMP signaling is related to development and pathogenicity [87,88]. We also found that 19 genes were mapped to terpenoid backbone biosynthesis (Figure 9), which involves terpene synthesis with the role of the cytochrome P450 gene [89]. In the *M. grisea* PMg_Dl genome, 127 Cytochrome P450 (CYP) families were identified, which perform a broad range of roles such as metabolic function and survival in the natural environment, with an impact on pathogenicity [90]. Cytochrome P450 is documented for the synthesis of various secondary metabolites for protection, immune suppressor, and mycotoxic responses [91]. Taken together, the genomic data resource of pearl millet-infecting *M. grisea* PMg_Dl would offer an opportunity for population genetic studies to understand the population biology of *M. grisea* infecting not only pearl millet but also other major and minor millets, as has been recently published for the rice blast pathogen, *Magnaporthe oryzae* [20,58].

## 5. Conclusions

In the present study, the hybrid de novo genome assembly and functional annotation of pearl millet-infecting *M. grisea* PMg_Dl were performed. Whole-genome sequence functional annotation revealed the presence of various CAZymes with high GH families and cell wall-degrading enzymes. We also observed various pathways of signaling, starch and sucrose metabolism, pentose and glucuronate interconversions, and secondary metabolism. The presence of various transposable elements, pathogenicity-related genes, and virulence factors was also documented. Among protein families, WD domain, cytochrome P450, major facilitator superfamily, and protein kinase domain were highly prevalent in this genome. The genome data provide a fundamental knowledge base for the cereal killer *Magnaporthe grisea*. The present report expands the toolbox for biological and genomic studies concerning the fungal pathogen in general and *Magnaporthe* in particular. 

## Figures and Tables

**Figure 1 jof-08-00614-f001:**
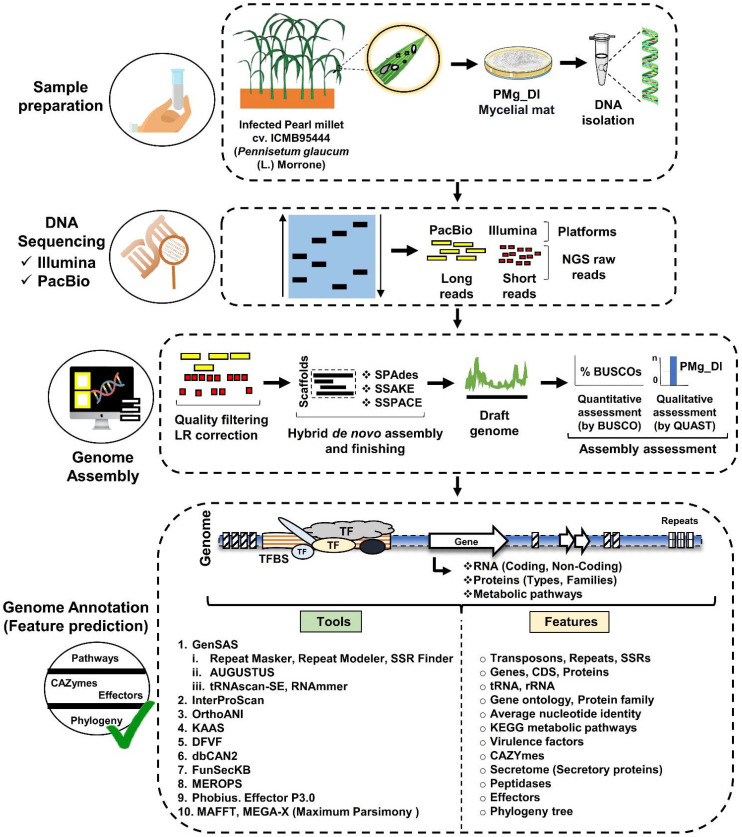
Schematic illustration of WGS analysis pipeline used for revealing versatile metabolic pathways, protein families, and virulence factors in *M. grisea* strain PMg_Dl. The entire pipeline was divided into four sequential major steps (Sample preparation, DNA sequencing, Genome Assembly, and Genome annotation). In the first step, the fungus was isolated from peal millet cultivar ICMB95444 showing blast lesion symptoms using a modified spore drop method. Following the isolation, the fungal genomic DNA was extracted. In the second step, the fungal DNA was subjected to sequencing library preparation followed by Illumina NextSeq 500 (2 × 150 bp chemistry) and PacBio RS II platform (P6-C4 chemistry) for generating short and long reads, respectively. In the third step, the draft genome was assembled using a series of tools and software after quality filtering. In the last step, functional genome annotation was performed to predict several characteristics of the *M. grisea* genome such as tRNA and rRNA presence, simple sequence repeats superfamily, signal peptides, gene ontology, metabolic pathways, CAZymes, secretory proteins, proteolytic proteins, potential effectors, and virulence factors. In addition, the phylogeny of *M. grisea* strain PMg_Dl was also studied.

**Figure 2 jof-08-00614-f002:**
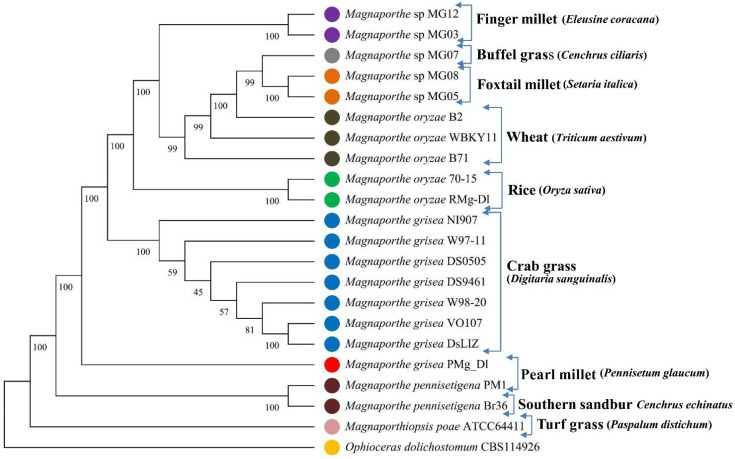
Phylogenetic tree of single-copy orthologous (SCO) proteins of 20 species belonging to *Magnaporthe* genera. The other two *Magnaporthiopsis poae* ATCC64411 from *Magnaporthaceae* and *Ophioceras dolichostomum* CBS114926 from *Ophioceraceae* were taken as an outgroup. The tree was generated using the concatenated 100 SCOs commons in all 22 species through maximum parsimony phylogeny with 1000 bootstrap method and search using subtree–prune–regraft (SPR), totaling 237,605 covered sites.

**Figure 3 jof-08-00614-f003:**
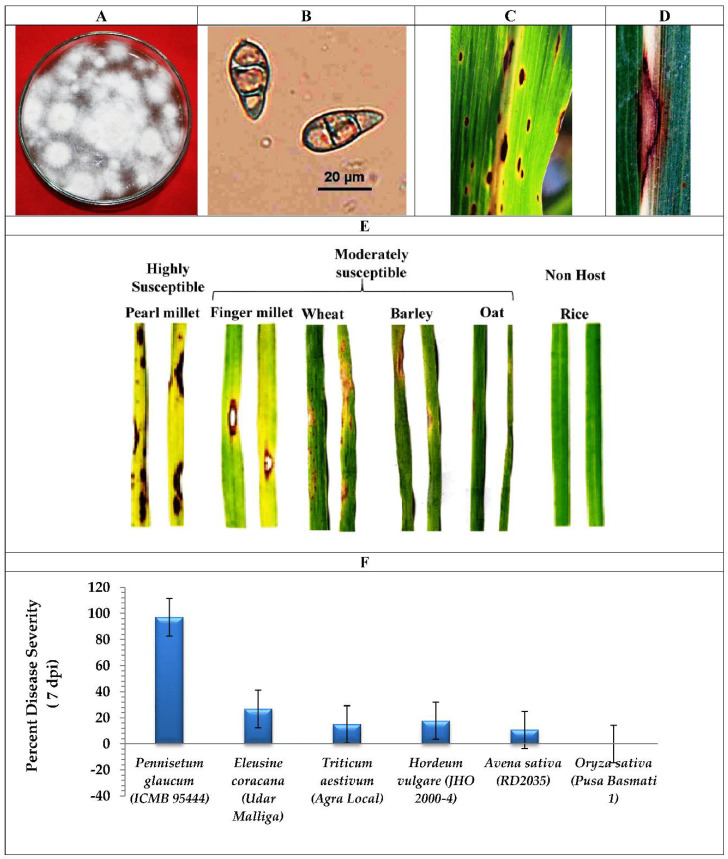
Mycelial and conidial features, pathogenicity, and host range of *Magnaporthe grisea* PMg_Dl. (**A**) Complete mycelial growth of *M. grisea* PMg_Dl showing sporulation as gray coloration. (**B**) Microscopic image showing three-celled conidia at 400 X magnification. (**C**) Blast lesions on pearl millet leaf in the field. (**D**) Close-up images of necrotic lesion on pearl millet leaf. (**E**) Pathogenicity assay showing high susceptibility of pearl millet to *M. grisea* PMg_Dl; finger millets, wheat, barley, and oat showed moderate susceptibility; rice is not a host to *M. grisea* PMg_Dl. (**F**) Blast severity score recorded on various Poaceae members; bar represents standard error of the mean difference.

**Figure 4 jof-08-00614-f004:**
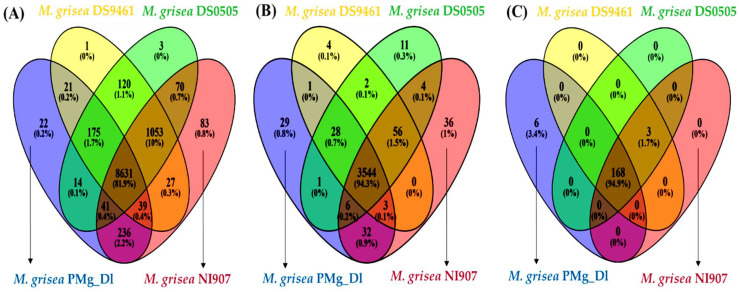
The predicted proteins of *M. grisea* PMg_Dl and other genomes’ comparative annotation. (**A**) Orthologous genes, (**B**) Protein family (PFAM), and (**C**) CAZymes. In (**C**), the CAZyme families such as AA1, GH109, GH35, GT109, GT61, and PL42 were found uniquely present in *M. grisea* PMg_Dl, whereas other families, namely, CBM67, GH145, and GH65 were found to be shared among DS9461, DS0505, and NI907genomes.

**Figure 5 jof-08-00614-f005:**
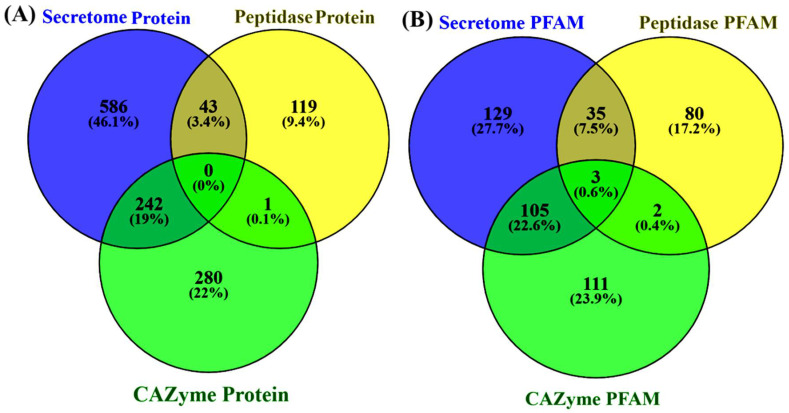
The identified proteins were classified against secretome, MEROPS (peptidase), CAZymes, and PFAM database. (**A**) The protein sequences shared and uniquely classified into secretome, MEROPS, and CAZymes. (**B**) The protein sequences classified against each database (secretome, MEROPS, CAZymes) mapped to distinct PFAM, thus depicting the shared and uniquely PFAM in mapped sequences.

**Figure 6 jof-08-00614-f006:**
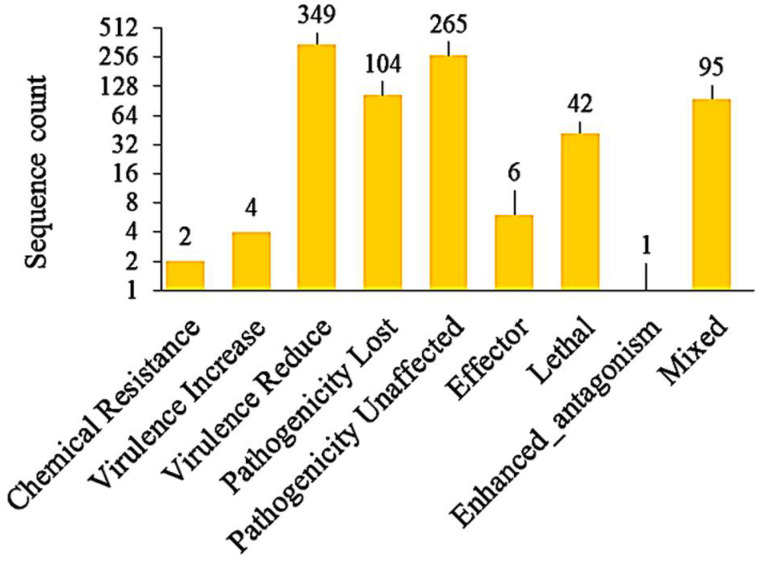
Identified pathogen–host interaction categories of *M. grisea* PMg_Dl-encoded protein sequences.

**Figure 7 jof-08-00614-f007:**
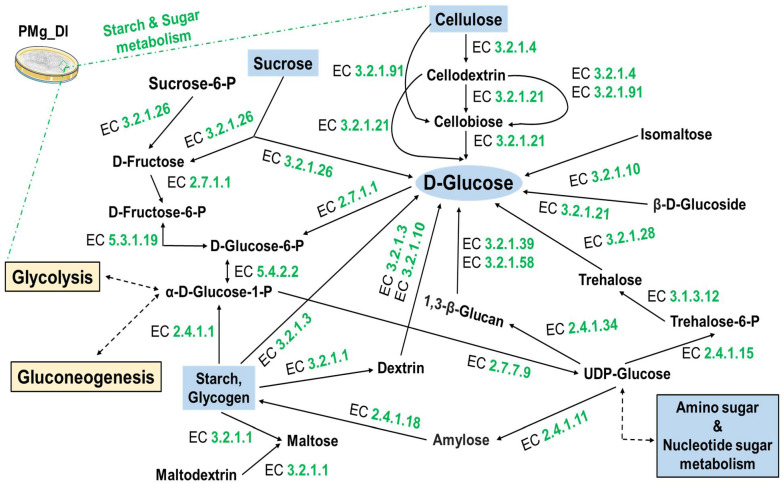
Schematic illustration of starch and sucrose metabolism pathways showing the cellulose polysaccharide breakdown routes and main cell wall degradation enzymes (CWDEs), namely, EC 3.2.1.4, EC 3.2.1.91, and 3.2.1.21. The sysnames of enzymes related to the biosynthetic machinery are depicted in purple-colored EC numbers: EC 3.2.1.4: 4-β-D-glucan 4-glucanohydrolase; EC 3.2.1.21: β-D-glucoside glucohydrolase; EC 3.2.1.91: 4-β-D-glucan cellobiohydrolase (non-reducing end); EC 3.2.1.26: β-D-fructofuranosidefructohydrolase; EC 2.7.1.1: ATP:D-hexose 6-phosphotransferase; EC 5.3.1.9: D-glucose-6-phosphate aldose-ketose-isomerase; EC 2.4.1.15: UDP-α-D-glucose: D-glucose-6-phosphate 1-α-D-glucosyltransferase; EC 3.1.3.12: α, α-trehalose-6-phosphate phosphohydrolase; EC 3.2.1.28: α, α-trehaloseglucohydrolase; EC 2.4.1.11: UDP-α-D-glucose: glycogen 4-α-D-glucosyltransferase; EC 2.4.1.18: (1->4)-α-D-glucan: (1->4)-α-D-glucan 6-α-D-[(1->4)-α-D-glucano]-transferase; EC 2.4.1.1: (1->4)-α-D-glucan:phosphateα-D-glucosyltransferase; EC 3.2.1.1: 4-α-D-glucan glucanohydrolase; EC 3.2.1.3: 4-α-D-glucan glucohydrolase; EC 3.2.1.10: Oligosaccharide 6-α-glucohydrolase; EC 5.4.2.2: α-D-glucose 1,6-phosphomutase; EC 2.4.1.34: UDP-glucose:(1->3)-β-D-glucan 3-β-D-glucosyltransferase; EC 3.2.1.39: 3-β-D-glucan glucanohydrolase; EC 3.2.1.58: 3-β-D-glucan glucohydrolase; EC 3.2.1.21: β-D-glucoside glucohydrolase and EC 2.7.7.9: UTP: α-D-glucose-1-phosphate uridylyltransferase.

**Figure 8 jof-08-00614-f008:**
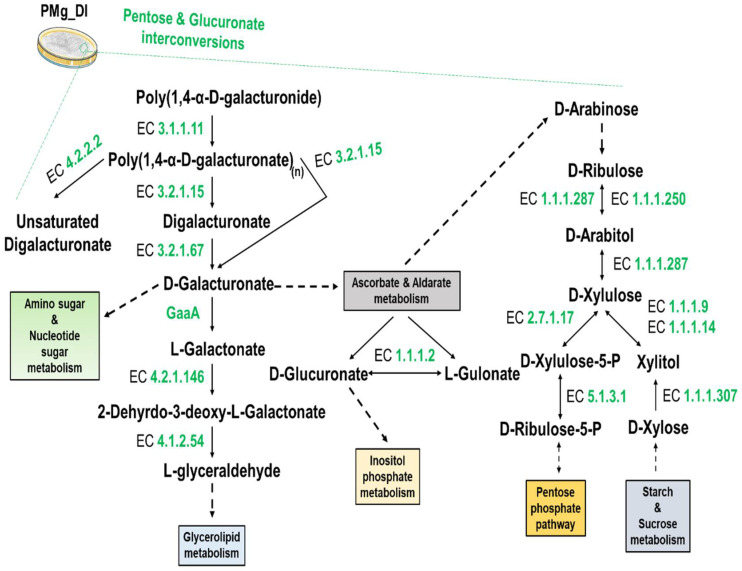
Schematic illustration of pentose and glucoronate interconversion pathways showing the pectin and xylan breakdown routes and main enzymes. The sysnames of enzymes related to the biosynthetic machinery are depicted in green-colored EC numbers: EC3.1.1.11: Pectin pectylhydrolase; EC3.2.1.15: (1->4)-α-D-galacturonan glycanohydrolase (endo-cleaving); EC3.2.1.67: Poly[(1->4)-α-D-galacturonide] non-reducing-end galacturonohydrolase; EC3.2.1.15: (1->4)-α-D-galacturonan glycanohydrolase (endo-cleaving); EC4.2.2.2: ((1->4)-α-D-galacturonan lyase; EC4.2.1.146: L-galactonate hydro-lyase (2-dehydro-3-deoxy-L-galactonate-forming); EC4.1.2.54: 2-dehydro-3-deoxy-L-galactonate L-glyceraldehyde-lyase (pyruvate-forming); EC1.1.1.287: D-arabinitol: NADP^+^ oxidoreductase; EC1.1.1.250: D-arabinitol: NAD^+^ 2-oxidoreductase (D-ribulose-forming); EC2.7.1.17: ATP:D-xylulose 5-phosphotransferase; EC1.1.1.9: Xylitol: NAD^+^ 2-oxidoreductase (D-xylulose-forming); EC1.1.1.14: L-iditol: NAD^+^ 2-oxidoreductase; EC1.1.1.2: Alcohol:NADP^+^ oxidoreductase; EC5.1.3.1: D-ribulose-5-phosphate 3-epimerase; EC1.1.1.307: Xylitol: NAD(P)^+^ oxidoreductase and GaaA: (D-galacturonate reductase) (EC:1.1.1).

**Figure 9 jof-08-00614-f009:**
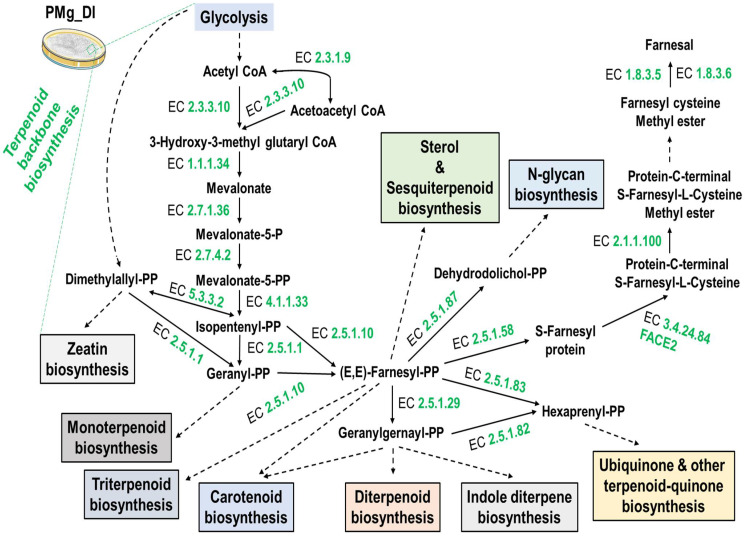
Schematic illustration of terpenoid backbone biosynthesis pathways of *M. grisea* PMg_D1-associated natural secondary metabolites synthesis and linked downstream pathways. The sysnames of enzymes related to the biosynthetic machinery are depicted in green-colored EC numbers: EC 2.3.1.9: Acetyl-CoA: acetyl-CoA C-acetyltransferase; EC 2.3.3.10: Acetyl-CoA: acetoacetyl-CoA C-acetyltransferase (thioester-hydrolysing, carboxymethyl-forming); EC 1.1.1.34: (R)-mevalonate: NADP^+^ oxidoreductase (CoA-acylating); EC 2.7.1.36: ATP: (R)-mevalonate 5-phosphotransferase; EC 2.7.4.2: ATP: (R)-5-phosphomevalonate phosphotransferase; EC 5.3.3.2: Isopentenyl-diphosphate δ3-δ2-isomerase; EC 4.1.1.33: ATP: (R)-5-diphosphomevalonate carboxy-lyase (adding ATP; isopentenyl-diphosphate-forming); EC 2.5.1.1: Dimethylallyl-diphosphate:isopentenyl-diphosphate dimethylallyltranstransferase; EC 2.5.1.10: Geranyl-diphosphate:isopentenyl-diphosphategeranyltranstransferase; EC 2.5.1.87: (2E,6E)-farnesyl-diphosphate:isopentenyl-diphosphatecistransferase (adding 10--55 isopentenyl units); EC 2.5.1.58: Farnesyl-diphosphate:protein-cysteine farnesyltransferase; EC 2.5.1.83: (2E,6E)-farnesyl-diphosphate:isopentenyl-diphosphatefarnesyltranstransferase (adding 3 isopentenyl units); EC 2.5.1.29: (2E,6E)-farnesyl-diphosphate: isopentenyl-diphosphatefarnesyltranstransferase; EC 2.5.1.82: Geranylgeranyl-diphosphate:isopentenyl-diphosphate transferase (adding 2 isopentenyl units); EC 3.4.24.84: STE24 endopeptidase; EC 2.1.1.100: S-adenosyl-L-methionine: protein-C-terminal-S-farnesyl-L-cysteine O-methyltransferase; EC 1.8.3.5: S-prenyl-L-cysteine:oxygen oxidoreductase; EC 1.8.3.6: S-(2E,6E)-farnesyl-L-cysteine oxidase and FACE2: Prenyl protein peptidase.

**Table 1 jof-08-00614-t001:** General features of the assembled whole genome of *M. grisea* PMg_Dl.

Attributes	*M. grisea* PMg_Dl
Illumina NextSeq 500 PE	43,962,401 (PE reads); 13.1 Gb
Illumina NextSeq 500 MP	17,160,010 (MP reads); 3.4 Gb
PacBio RS II	148,768 single end; 1.1 Gb
Genome size	47,897,363
Number of scaffolds	341
Scaffold N50	765,468
Largest scaffold	2,246,534
GC%	47.3
No. of genes	10,218
Proteins	10,184
rRNA	38
tRNA	209
SSR	79,317
SSR bases	243,245
Virulence factors	51
CAZymes	539
Peptidase	163
Secretome proteins	871
Effector proteins	594

**Table 2 jof-08-00614-t002:** Detected transposable elements in *M. grisea* PMg_Dl genome.

Family	Copy Number *(M. grisea* PMg_Dl)
DNA	326
DNA/TcMar-Fot1	471
DNA/TcMar-Mariner	38
DNA/TcMar-Tc1	20
DNA/hAT-Ac	54
DNA/hAT-Restless	54
Type: DNA	971
LINE/Tad1	147
Type: LINE	173
LTR/Copia	1672
LTR/Gypsy	837
Type: LTR	2509
Type: EVERYTHING_TE	3653
Type: Simple_repeat	28
Type: Unknown	7033
LINE/CRE-Cnl1	26
DNA/TcMar-Ant1	8
Total	18,020

**Table 3 jof-08-00614-t003:** Comparative CAZyme family profile in *M. grisea* PMg_Dl, *P. oryzae* DS9461, *M. oryzae* DS0505, and *M. oryzae* NI907genomes.

Family	*M. grisea* PMg_Dl *	*M. grisea* DS9461 *	*M. grisea* DS0505 *	*M. grisea* NI907 *
AA	127	123	121	117
CBM	11	15	15	15
CE	50	48	50	51
GH	253	256	254	252
GT	91	95	94	96
PL	7	6	6	6
Total	539	543	540	537

* The genomes’ GenBank assembly accession: *M. grisea* PMg_Dl (GCA_003933175.1), *M. grisea* DS9461 (GCA_001548795.1), *M. grisea* DS0505 (GCA_001548815.1), *M. grisea* NI907 (GCA_004355905.1). The abbreviations of CAZyme families: AA; auxiliary activity, CBM; carbohydrate-binding module, GH; glycoside hydrolase, GT; glycosyltransferase, PL; polysaccharide lyases.

**Table 4 jof-08-00614-t004:** Carbohydrate metabolism KEGG (Kyoto Encyclopedia of Genes and Genomes) pathway for *M. grisea* PMg_Dl.

Carbohydrate Metabolism Pathways	*M. grisea* PMg_Dl (Gene Count)
KO:00010 Glycolysis/Gluconeogenesis	25
KO:00020 Citrate cycle	20
KO:00030 Pentose phosphate pathway	18
KO:00040 Pentose and glucuronate inter conversions	18
KO:00051 Fructose and mannose metabolism	21
KO:00052 Galactose metabolism	14
KO:00053 Ascorbate and aldarate metabolism	6
KO:00500 Starch and sucrose metabolism	26
KO:00520 Amino sugar and nucleotide sugar metabolism	26
KO:00620 Pyruvate metabolism	28
KO:00630 Glyoxylate and dicarboxylate metabolism	21
KO:00650 Butanoate metabolism	12
KO:00640 Propanoate metabolism	18
KO:00660 C5-Branched dibasic acid metabolism	3
KO:00562 Inositol phosphate metabolism	21
Total	277

**Table 5 jof-08-00614-t005:** Signal transduction KEGG (Kyoto Encyclopedia of Genes and Genomes) pathways for *M. grisea* PMg_Dl.

Signal Transduction Pathways	*M. grisea* PMg_Dl (Gene Count)
KO:02020 Two-component system	19
KO:04014 Ras signaling pathway	19
KO:04015 Rap1 signaling pathway	10
KO:04010 MAPK signaling pathway	17
KO:04013 MAPK signaling pathway—fly	14
KO:04016 MAPK signaling pathway—plant	4
KO:04011 MAPK signaling pathway—yeast	56
KO:04012 ErbB signaling pathway	6
KO:04310 Wnt signaling pathway	13
KO:04330 Notch signaling pathway	4
KO:04340 Hedgehog signaling pathway	5
KO:04341 Hedgehog signaling pathway—fly	7
KO:04350 TGF-beta signaling pathway	7
KO:04390 Hippo signaling pathway	9
KO:04391 Hippo signaling pathway—fly	7
KO:04392 Hippo signaling pathway—multiple species	6
KO:04370 VEGF signaling pathway	9
KO:04371 Apelin signaling pathway	14
KO:04630 Jak-STAT signaling pathway	4
KO:04064 NF-kappa B signaling pathway	5
KO:04668 TNF signaling pathway	3
KO:04066 HIF-1 signaling pathway	15
KO:04068 FoxO signaling pathway	14
KO:04020 Calcium signaling pathway	11
KO:04070 Phosphatidylinositol signaling system	18
KO:04072 Phospholipase D signaling pathway	13
KO:04071 Sphingolipid signaling pathway	20
KO:04024 cAMP signaling pathway	12
KO:04022 cGMP-PKG signaling pathway	11
KO:04151 PI3K-Akt signaling pathway	24
KO:04152 AMPK signaling pathway	23
KO:04150 mTOR signaling pathway	38
Total	437

## Data Availability

The genome assembly data of *M. grisea* PMg_Dl (Illumina and PacBio RSII) have been submitted to the NCBI GenBank assembly with accession no. GCA_003933175.1, with raw reads accession SRR8573217, SRR8573216, andSRR8776454. The authors state that in the current study, the necessary data for the conclusion and Appendix A are included.

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
