# Peer review of "Structured Framework and Genome Analysis of *Magnaporthe grisea* Inciting Pearl Millet Blast Disease Reveals Versatile Metabolic Pathways, Protein Families, and Virulence Factors"

_jof, 2022, doi:10.3390/jof8060614_

Round 1

Reviewer 1 Report

In the MS “Structured framework and genome analysis of Magnaporthe 2 grisea inciting pearl millet blast disease reveals versatile meta-3 bolic pathways, protein families, and virulence factors”, authors demonstrated the genomic basis of the pathogenicity and underlying biochemical pathways in Magnaporthe using genome-sequence of a pearl millet infecting M. grisea PMg_Dl generated by dual NGS techniques, Illumina NextSeq 500 and PacBio RS II. The most obvious feeling is that the author only lists the data and does not connect these data with characteristics of M. grisea PMg_Dl.

  1. Can you get more information from the phylogenetic tree?
  2. Figure 3. Is the picture narrowed?
  3. Analysis of Orthologous Genes, Protein Family, and CAZymes in Assembled Genomes. Are these genes related to the specificity of grisea PMg_Dl?
  4. How many chromosomes grisea PMg_Dl.I has?
  5. The MS focused on the conserved information of Magnaporthe. It is better to analyze these data to give some information about the host specificity.
  6. Please provide some pictures of the colony morphology, infected structure, and pathogenicity of grisea PMg_Dl.I.

Reviewer 2 Report

Reddy et al. conducted a pretty comprehensive study on Magnaporthe grisea and the manuscript was written with a good presentation. 

Just some minor comments to further improve the MS:

  1. the information in the legend of Figure 1 is not sufficient and clear. The authors should break the diagram into different steps and describe each step in detail in the legend.
  2. Line 118, the BUSCO version, is not indicated. 
  3. The authors can provide another Supp table, by comparing the assembly quality (N50 or BUSCO) genome they generated and other published genomes listed in Figure 2. This information can be useful to assess the quality of the genome and how it has been improved. 
  4. The content of the abstract is quite redundant. Can remove some unnecessary text and only keep those most significant findings and content/ 

Round 2

Reviewer 1 Report

Comments are answered